# Analysis of Factors Relevant to Revenue Enhancement in Hernia Interventions (SwissDRG G09)

**DOI:** 10.3390/healthcare9070862

**Published:** 2021-07-08

**Authors:** Bassey Enodien, Stephanie Taha-Mehlitz, Marta Bachmann, Victor E. Staartjes, Maike Gripp, Tobias Staudner, Anas Taha, Daniel Frey

**Affiliations:** 1Department of Surgery, GZO Hospital, 8620 Wetzikon, Switzerland; bassey.enodien@gzo.ch (B.E.); Marta.bachmann@GZO.ch (M.B.); Maike.Gripp@GZO.ch (M.G.); Daniel.Frey@GZO.ch (D.F.); 2Clarunis University Center for Gastrointestinal and Liver Diseases, 4002 Basel, Switzerland; stephanie.taha@clarunis.ch; 3Independent Researcher, 8005 Zurich, Switzerland; victor.staartjes@gmail.com; 4Department of Visceral and Thoracic Surgery Cantonal Hospital Winterthur, 8400 Winterthur, Switzerland; Tobias.Staudner@fau.de; 5Faculty of Medicine, University of Basel, 4001 Basel, Switzerland

**Keywords:** costs, economy, EBITDA, SwissDRG, hernia, G09

## Abstract

Background: Since diagnosis-related groups (SwissDRG) were established in Switzerland in 2012, small and medium-size hospitals have encountered increasing financial troubles. Even though hernia repair operations are frequent, most hospitals fail to cover their costs with these procedures. Previous studies have focused mainly on analyzing costs and the contributing factors but less on variables that can be positively influenced. Therefore, this study aims to identify the relevant and influenceable factors for revenue growth in hernia repair surgery. Methods: Data from all patients who underwent the SwissDRG G09 surgery for a hernia in 2019 were analyzed. The contribution margin (CM4), as well as any over- or under-coverage, was correlated to case-specific costs. Results: A total of 168 patients received hernia repair surgery with the SwissDRG code G09. The average revenue/loss generated by one procedure was CHF −623.84. Procedures covered by the General Health Insurance (OKP) generated a loss of CHF −830.70 on average, whereas procedures covered by private insurance companies (VVG) generated revenue of CHF +1100 on average. Significant factors impacting the profitability of hernia repair operations were teaching during surgery (*p* < 0.005), the surgical operating time (*p* < 0.001), the total anesthesia time (*p* < 0.001), the number of surgeons present (*p* = 0.022), the insurance state of patients (*p* < 0.001), and the type of surgery (*p* < 0.01 for Lichtenstein’s procedure). Conclusions: This study reveals that hernia repair surgery performed under cost coverage by OKP is generally unprofitable. Our results further imply that the most important and influenceable factors for revenue enhancement are the quality and process optimization of the surgical department. To compensate for this deficit, hospitals should aim to increase the percentage of patients with private health insurance coverage in their procedures. Since outpatient surgery does not provide a valid alternative due to the low reimbursement by insurance companies, the cost efficiency of inpatient hernia repair needs to be increased by process optimization of the surgical department; for instance, by providing specialized hernia teams performing with shorter operation times and high quality.

## 1. Introduction

Inguinal hernia repair is a frequent operation in the daily routine of a surgeon. Inguinal hernias are classified according to the European Hernia Society into medial and lateral (M/L) [1], according to size into 1 (max. 1 finger), 2 (1–2 fingers) and 3 (3 fingers or more) [1], as well as into primary hernias (P) and recurrences (R) [1]. For the management of inguinal hernias, there are recommendations of the European Hernia Society [1], consistent with the international guidelines for inguinal hernia management [2]. Due to the lower recurrence rates, surgical techniques with a mesh are generally recommended [1,2]. In rare cases, treatment without a mesh is justified; in this case, Shouldice treatment should be performed [1,2]. With regard to inguinal hernia management with mesh repair, the open technique, according to Lichtenstein, as well as endoscopic techniques using total extraperitoneal preperitoneal and trans-abdominal preperitoneal mesh placement are considered equivalent and standard treatment [1,2]. Surgery is feasible under inpatient as well as outpatient conditions and standardization of surgical techniques is recommended [1,2]. Definition and classification allow standardization, which contributes to process optimization and quality improvement.

In the literature, there are several articles from different countries dealing with cost analysis about inguinal hernia repair [3,4,5,6]. These articles showed that factors relevant to costs were age, comorbidities, surgical technique, bilateral repair, emergency surgery for incarceration, and postoperative complications needing surgical treatment [5]. Some studies further showed that endoscopic hernia repair is cost-effective [3,4,5] or even more cost-effective than Lichtenstein open surgery and is preferable to this technique in terms of cost [3,4]. The knowledge of the costs is an essential to analyze and optimize a process. In the studies mentioned above, the factors that led to increased costs were analyzed, but not the contribution margins. After reviewing the available literature, we concluded that we should not only examine and present the costs of the individual factors but their influence on the contribution margin. Factors such as age, comorbidities, unilateral or bilateral operations, as well as the fact of an emergency operation are significant cost factors as shown above [5], but they cannot be changed beforehand. Our hypothesis is based on the assumption that there are relevant but influenceable factors for the recovery enhancement, which we would like to show with the help of our analysis. Raakow et al. focused their publication from Germany on the GermanDRG system, which also used the approach of analyzing the contribution margins [6]. Here, apart from complications, the operation time was considered to be the most important factor [6].

Regarding revenue enhancement, it can be seen that outpatient operations in the SwissDRG system generate at least 1.9 up to 3.2 times less revenue than inpatient operations. Therefore, outpatient inguinal hernia repair is considered unprofitable from the point of view of the service providers. The majority of Swiss hospitals have a public service mandate, which means that they have to treat all patients and cannot select patients based on economic considerations. In Switzerland, there is general insurance within the scope of the compulsory health care insurance (OKP), as well as semi-private (SP) and private (P) supplementary insurance within the scope of the Insurance Contract Act (VVG). In the area of OKP, revenues are limited in accordance with SwissDRG, as explained below. The supplementary insurances (SP/P) reimburse services at a higher level and without limits, resulting in higher revenues within the scope of the VVG.

Diagnosis-related groups in Switzerland (SwissDRG) have been modeled after the GermanDRG system and were established in 2012 [7]. The DRG system aims to compensate somatic hospital services in a standardized way. In accordance with the Swiss Federal Health Insurance Act (KVG), the compensation for inpatient hospital services is regulated on a uniform basis according to a case-based lump sum. These reforms were designed to increase the transparency of medical services and create competition among health care providers. This should serve as an indirect incentive to reduce costs in the medical system and significantly shorten hospital stays. However, the increased quality competition together with the extreme costs and the concomitant liberalization of the health care market has led to an increasingly tense financial situation for Swiss hospitals [8]. Indeed, the actual effects of DRGs on the health care system are a matter of ongoing debate. For instance, the targeted reduction in length of hospital stay was only achieved in part [9]. Even though in-hospital mortality was reduced [9], an increase in readmissions within 30 days could be observed [9].

The compensation that hospitals receive for each case according to SwissDRG depends on the cost weight of the procedure and the hospital-specific base rate [7]. The cost weight of a case can be assessed through the following factors: main diagnosis (G09 A-E for this study), secondary diagnoses, interventions, and the severity code. These factors are relevant beyond the daily challenges of a surgeon. Indeed, these factors directly impact the financial outcomes of surgical departments.

Diagnosis-related groups achieve far more than a diagnostic code. Grouping software (e.g., Grouper) is used to algorithmically assign cases to their respective DRG according to their main diagnosis, as well as according to several other factors [10] including secondary diagnoses, procedures, gender, age, admission type, type of discharge, length of hospital stay, duration of intubation, and length of stay at the intensive care unit.

Hernia repair procedures are among the most frequent surgical procedures and are highly standardized [11]. However, in most hospitals, they fail to return a profit [6]. Therefore, this study aims to investigate the factors that are detrimental to cost-efficiency and influenceable for the revenue enhancement in hernia repair surgery.

## 2. Materials and Methods

### 2.1. Statistical Analysis

Continuous variables are reported as means and standard deviations (SD), while categorical variables are reported as numbers and percentages. There were no missing data [12].

Welch’s two-sample t-test and Pearson’s Chi-Square test were used to compare the characteristics of patients with negative and positive contribution margins [13]. Welch’s two-sample test was applied to compare continuous variables among the two groups, while Pearson’s Chi-Square test was applied to compare categorical variables between groups. The optimal binary break-even points [14] of the surgery and anesthesia times were determined based on the area under the curve (AUC) and the “closest-to-(0, 1)-criterion” [15]. The effects of these times on the contribution margins were presented using “scatterplots with marginal histograms” and “Pearson correlation coefficients”. For the statistical analysis, the free software “R Version 4.0.5” (R Core Team, Vienna, Austria) of the R Project for statistical computing from the R Foundation was used [16]. The significance level was defined as *p* < 0.05.

### 2.2. Data Collection

Data from all the patients admitted to the hospital Wetzikon between January and December in 2019 for hernia repair interventions with the assigned DRG G09 and operated (electively or emergently) with the Swiss operational classification codes (CHOP) 53.07.21 (operation for inguinal hernia, laparoscopic with implant), 53.23.21 (operation for femoral hernia, laparoscopic with implant) and 53.06.21 (operation for inguinal hernia, surgical open with implant) were included in this study. The hospital Wetzikon is a private hospital with a public service contract.

### 2.3. Variables and Definition

Data on age, sex, body mass index (BMI), surgical operating time, total anesthesia time, ASA-classification, teaching status of the operation, insurance status, number of surgeons, unilateral or bilateral operations, experience level of the attending operators (CSu (Chief of Surgery)/CSe (Chief of Service)/AS (Attending Surgeon)/RS (Resident Surgeon)), and additional procedures during the operation were collected. The insurance status is either a compulsory basic care or semi-private and private hospital care. Differences are the coverage of extra services, which means the patient can choose the physician in the hospital and is entitled to a single bedroom (private) or double bedroom (semi-private).

A structured overview of the variables relevant for this analysis is presented in Table 1. These variables were correlated with the contribution margin (CM4) [17] of the individual procedures, which could be obtained from the controlling department’s internal data processing system. The CM4 value indicated a possible over or under-coverage in relation to the case-specific costs [17]. The base price of a DRG case-based lump sum is calculated by multiplying the respective evaluation ratio by the base case rate [18]. In 2019, the base case rate was CHF 9650.

## 3. Results

In 2019, 168 patients were operated on in our hospital for hernia under the SwissDRG code G09. The average age was 61.73 years (range: 19–94 years). In total, 12.5% of the patients were women and 87.5% were men, and 133 endoscopic total extraperitoneal hernioplasty procedures (115 bilateral/18 unilateral) and 35 open procedures according to Lichtenstein (14 bilateral/21 unilateral) were performed.

The mean operating time was 80.47 min (range: 25–193 min), and the mean anesthesia time was 155.48 min (range: 61–284 min). The combined mean induction and emergence time was 75.01 min.

Postoperative complications occurred in seven patients (4.19%). The mean length of hospital stay was 2.13 days (range: 1–13 days). See also Table 2.

The 168 surgeries were additionally broken down in terms of insurance status and profitability. A mean deficit of CHF −623.84 per case (range: CHF −11,182 to +5308) occurred across all insurance classes. Hernia interventions in the scope of OKP (general health insurance) were unprofitable, with a contribution margin of CHF −830.70 per case. Hernia interventions in the scope of VVG (private and semi-private insurance) were profitable, with a contribution margin of CHF +1100 per case.

As described in our hypothesis in the introduction and confirming the conclusions of the existing literature [6], our analysis showed that age, gender, BMI, ASA classification, previous surgeries, unilateral or bilateral surgeries, and the experience of the main surgeon did not have a significant influence on the cost contribution [6]. These factors cause costs but cannot be influenced and are consequently not relevant for revenue enhancement and our analysis [4], since in the Swiss public health care system, public service contracts are issued that obligate the care of all patients. In our hospital, being a private hospital with a public service contract, patient selection according to the health insurance status is not legal.

The significant factors influencing the contribution margin were teaching during surgery (*p* < 0.005) (Table 2), the surgical operating time (*p* < 0.001) (Table 2, Figure 1), the total anesthesia time (induction, surgical time, and emergence) (*p* < 0.001) (Table 2, Figure 2), the number of surgeons present (*p* = 0.022) (Table 2), the patient’s insurance status (*p* < 0.001) (Table 2), and the surgical technique (*p* < 0.01 for Lichtenstein’s operation) (Table 2). Pure anesthesia time (induction and emergence) (Table 2, Figure 3) was not significant (*p* = 0.108).

Our results confirm the hypothesis that there are patient-independent factors influencing revenue enhancement. It should be noted that the surgical time as a component of the total anesthesia time influences it. The pure anesthesia time as described above is not a significant factor. We could show that profitable hernia care is possible if the surgical time is kept below the break-even point of 73.5 min (Figure 1), and for surgical time and anesthesia time below 145.5 min (Figure 3). Looking at the literature, our results confirm this for the scope area of SwissDRG [6]. In addition to the existing literature, our analysis shows that the number of surgeons present, the teaching status of the operation is relevant as well as the insurance status. The choice of surgical technique is relevant in our analysis and confirms recommendations in the literature regarding the preferred endoscopic technique as well as the cost [1,2,3,4,5].

In addition, the break-even points [10] for the individual procedural steps as well as the entire process were calculated. For the surgical operating time, the break-even point was at 73.5 min (Figure 4). For the pure anesthesia time (induction and emergence) and total process time (total anesthesia time), the break-even points were 74.5 min (Figure 5) and 145.5 min (Figure 6), respectively.

## 4. Discussion

Comparable and transparent services as well as competitiveness among providers were stated as objectives of the introduction of SwissDRG [7]. Since the revision of the KVG with the introduction of SwissDRG and the concomitant liberalization of the health care market, quality competition coupled with extreme cost pressure has led to an increasingly tense financial situation for Swiss hospitals. PricewaterhouseCoopers published the study “Swiss hospitals: This is how healthy their finances were 2019” (“Schweizer Spitäler: so gesund waren die Finanzen 2019”) [8] on this topic in November 2020. This study revealed that a large number of Swiss hospitals experience serious deficits. The current pandemic is further pushing Swiss hospitals to their limits, both medically and economically. Three-quarters of large Swiss full-service hospitals and university hospitals were expected to experience a deficit in 2020 [19]. Up to 65% of all Swiss hospitals were expected to experience a deficit in 2020 [19]. Of 150 large Swiss hospitals surveyed, 91% expected the situation to deteriorate further over the next 5 years [19]. Numerous regulatory interventions and binding obligations are increasingly narrowing hospitals’ scope of action. Consequently, contribution margins must be increasingly analyzed and differentiated by individual service areas, and additional insights derived from the analysis must be implemented in hospital management [20].

For senior physicians (“Kaderärzte”), this means assuming responsibility for their departments’ cost structure and revenues in addition to clinical, personnel, and continuing educational duties [21]. The annually changing dynamic SwissDRG reimbursement systems, as well as highly standardized procedures, at annually decreasing cost weights of high-profit procedures and thus decreasing reimbursement contributions additionally limit the opportunity to increase revenues [20]. Thus, since the revision of the KVG and the introduction of SwissDRG in 2012, there has been an increasing pressure for more economical service provisions and a need for an increase in the number of cases in strategically important areas, as was already the case in Germany [22]. An important key figure in this context is the EBITDA (Earnings before Interests, Taxes, Depreciation and Amortization) margin. The EBITDA margin is one of the key factors considered in the context of company analysis as it shows the ratio of EBITDA to total revenue for hospitals and thus is a valuable indicator for successful business operations [23]. The hospital must use its EBITDA to finance depreciation, amortization, and capital cost in particular. Therefore, the EBITDA margin reveals the ability of a hospital to refinance these costs [23] and thus provides an insight into the financial health of a company/hospital. According to the study by PricewaterhouseCoopers mentioned above, only a few hospitals achieve the important goal of financial health defined by EBITDA margin of at least 10% [8]. Indeed, EBITDA margins for Swiss hospitals have even been declining on average in recent years (2016–2019) [8]. Against this background, it is important to analyze and optimize areas that are running at a loss in order to achieve an EBITDA margin of at least 10% and thereby maintain financially healthy hospitals.

To achieve the required profit and the above-mentioned EBITDA margin of preferably over 10% while maintaining the service mission assigned to Swiss hospitals, the comprehensive control and monitoring of the clinics’ cost and revenue structures is indispensable [20].

Therefore, the question arises regarding under which conditions inpatient hernia interventions can contribute to positive departmental and hospital performance. To distinguish profitable from non-profitable cases within the SwissDRG system, knowledge of the contribution margins of the individual SwissDRGs is essential [24]. The contribution margin indicates the share of a case group or an individual case in the coverage of fixed costs in the business [25]. The analysis of the SwissDRG G09 for the year 2019 (SwissDRG 8.0) showed that, in 2019, a large part (*n* = 115/68.45%) of these inpatient surgeries with SwissDRG code G09 (*n* = 168) did not return a profit for the hospital, whereas only a minority (*n* = 53/31.55%) were profitable. On average, hernia repair procedures led to a deficit of CHF −623.84 per case (contribution margin/DB4). Given the aforementioned high cost pressure in Switzerland’s health care system, this is a conceivably unfavorable result.

Therefore, we analyzed all the hernia repair surgeries within SwissDRG code G09 in our medium-sized hospital under inpatient conditions in 2019 and identified factors that led to interventions failing to cover their cost. Our results confirmed the assumption that, in the SwissDRG regime, in a very similar manner to the situation in the German DRG system, hernia repair interventions on average fail to create positive revenues (CHF −623.84 loss per case), even though the interventions are very frequent and highly standardized. Thus, achieving cost-effective hernia interventions is a very challenging task.

In the case of patients with supplementary insurance (semi-private and private), this is easier to achieve (CHF 1100 profit per case), as there is a higher underlying reimbursement (VVG) than in the case of patients with general insurance (OKP) (CHF −830.70 loss per case). Therefore, one way to improve profitability must be to increase the proportion of patients with supplementary insurance to the greatest extent possible. In 2019, 10.78% cases in the SwissDRG G09 were covered by the VVG. Furthermore, cost-effectiveness in cases of patients with general health insurance (OKP) must be improved. Our study revealed several factors that seem to be critical for maintaining positive contribution margins. These factors are teaching during surgery (*p* < 0.005) (Table 2), surgical operating time *p* < 0.001 (Table 2, Figure 1), total anesthesia time (induction, surgery time and emergence) (*p* < 0.001) (Table 2, Figure 2), number of surgeons present (*p* = 0.022) (Table 2), and surgical technique (*p* < 0.01 for Lichtenstein operation) (Table 2). In the case of relapse surgery, surgeons with a sufficiently high case load should be assigned [2], because they should be able to meet quality standards regarding operating time and complications even in difficult relapse cases. Complications must be minimized as much as realistically achievable to generate a profit. We did not analyze the processes of positioning care and the total exchange times. From our point of view, these processes will have to be optimized as well [26] in order to achieve cost-effective hernia repair interventions (SwissDrg G09). Since our findings can be applied to other DRG items, we believe our results to be of relevance for hospital controlling procedures. Future prospective and randomized studies will be necessary to implement our findings and to identify further subsets of surgery-related factors that influence profitability.

A further measure that does not directly increase revenues but allows for better cost control by the hospital and a more cost-effective use of staff would be the assignment of cost weights to surgeons [27]. This internally assigned cost weight would contribute to a more realistic representation of the real cost and to a more targeted control of personnel deployment. In practice, a chief of surgery (Chefarzt) with a correspondingly higher salary would receive a cost weight of, for example, 2.5, and a chief of service (Leitender Arzt) could be assigned a 1.5, while attending surgeons (Oberarzt), resident surgeons, and students would receive cost weights of 1, 0.8, and 0.6, respectively. This differentiation of personnel costs would allow for a better depiction of internal cost [27]. In a second step, insights gained from this analysis could be put to use to subsequently reduce personnel cost. Even though this measure does not directly increase revenue, the reduced cost would indirectly lead to increased profits. In addition, increased surgical quality also leads to increased profit as it minimizes complications and shortens operating times [2].

Recent debates have focused on the ongoing controversy between inpatient and outpatient hernia surgery. Although it is generally recommended by the European guideline for the care of inguinal hernias [1,2,6], outpatient surgery is not desirable for health care providers. Indeed, outpatient surgery in our example would be even less profitable than the aforementioned loss of CHF −623.84 per case in the inpatient setting. This is due to the fact that outpatient interventions are charged according to the tariff for outpatient medical services in Switzerland (Tarmed). By contrast, the reimbursement for inpatient interventions is covered by the general health insurance (OKP) and is about 1.9 times higher than for outpatient interventions [28]. The reimbursement for inpatient interventions covered by supplementary insurance (SP/P) within the scope of the Insurance Contract Act (VVG) is up to 3.2 times higher than for outpatient interventions [28]. A study from a German center revealed a loss of revenue of one third when 50% of inguinal hernia operations were switched to outpatient care [6,29]. Consequently, the factors revealed as significant for cost efficiency in this study need to be applied even more rigorously in the outpatient setting, as the already scarce reimbursement is even lower for outpatient procedures.

The main limitation of this analysis was the relatively low sample size, and this being a single-center analysis reflecting the economic issue surgery is facing due to the implementation of the DRG system. Prospective multicenter investigations or meta-analyses are likely to overcome this study design’s shortcomings and help generalize results.

Furthermore, in our investigation, we were only focusing on the surgical process of groin hernia repair. It is not clear which impact this critical topic of revenue enhancement to other areas of surgery has. Future investigations should also shed light on anesthesiological processes and post-surgery nursing areas to address enhancement in cost coverage.

## 5. Conclusions

Although hernia interventions are among the most frequent surgical procedures and are performed in a highly standardized manner, they fail to return positive contribution margins in the scope of the OKP. Our results with a mean deficit of CHF −623.84 per case underline this deficiency. Due to both the educational and service assignment of Swiss hospitals, the discontinuation of hernia interventions is unrealistic. In addition, the demand for outpatient hernia interventions has been increasing in recent years. There is also increasing pressure for quality assurance with certifications and the planned linking of service contracts and remuneration to quality standards. Thus, even greater efforts will be necessary to achieve profitability because reimbursement is even lower in the Tarmed tariff. Therefore, the existing processes must be intensively optimized to achieve at least cost coverage. Our results show that in hernia repair surgery, the factors “operating time” and “number of surgeons present” could be used for revenue enhancement through quality improvement, for instance, by forming specialized hernia surgery teams, which perform within short operating time, below the break-even point of 73.5 min but with high quality. Serving high quality with appropriate certification might also attract higher proportions of privately insured patients, enhancing profitability. To reduce the factor combined “surgery time and anesthesia time” close teamwork between the disciplines and specialized fast-track lines might optimize the performance and keep it below the break-even point of 145.5 min. Besides all efforts to enhance performance in the theatre, the controversy of surgical teaching must have been addressed as well. Since hernia surgery is a classical teaching procedure, future optimization processes should include this consideration to keep educating the future generation of young surgeons.

The necessary revenue enhancement with increasing demands on quality makes revenue enhancement by quality improvement and process optimization a crucial and up-to-date research topic in surgery.

## Figures and Tables

**Figure 1 healthcare-09-00862-f001:**
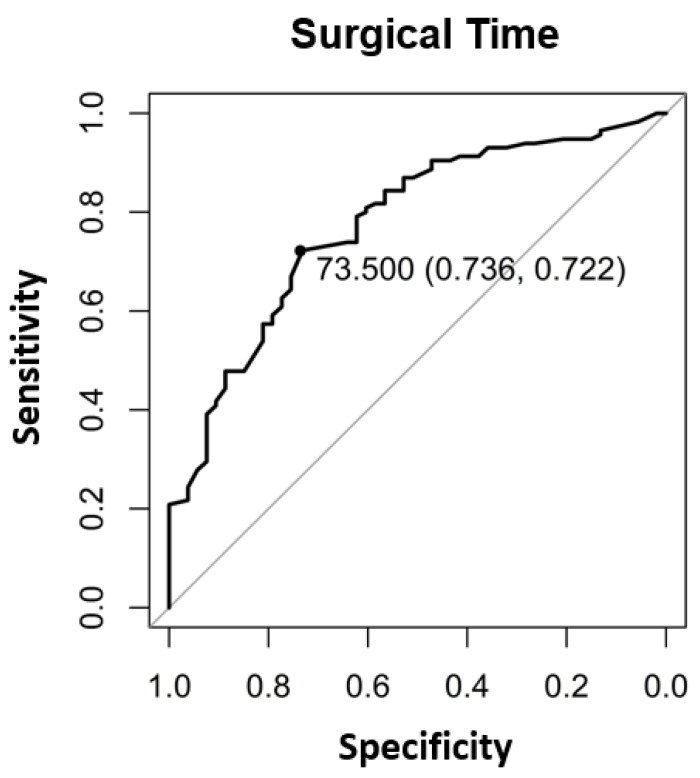
The binary break-even point for surgical time was 73.5 min determined with the Area under the Curve (AUC) and the “closest-to-(0, 1)-criterion”.

**Figure 2 healthcare-09-00862-f002:**
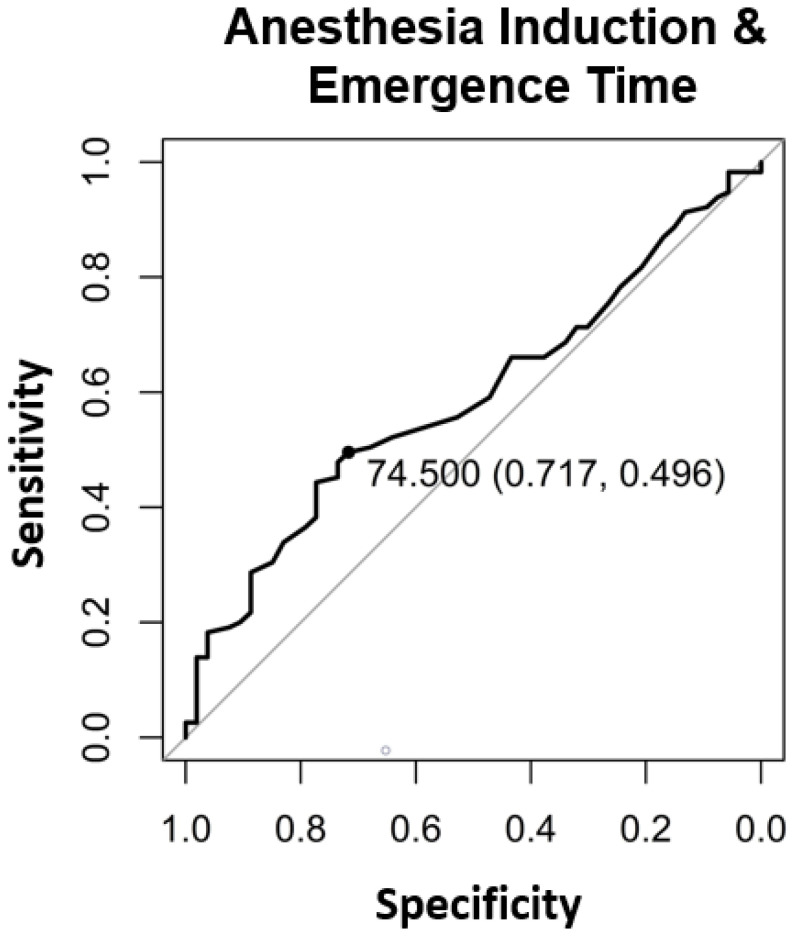
The binary break-even point for anesthesia and emergence time was 74.5 min determined with Area under the Curve (AUC) and the “closest-to-(0, 1)-criterion”.

**Figure 3 healthcare-09-00862-f003:**
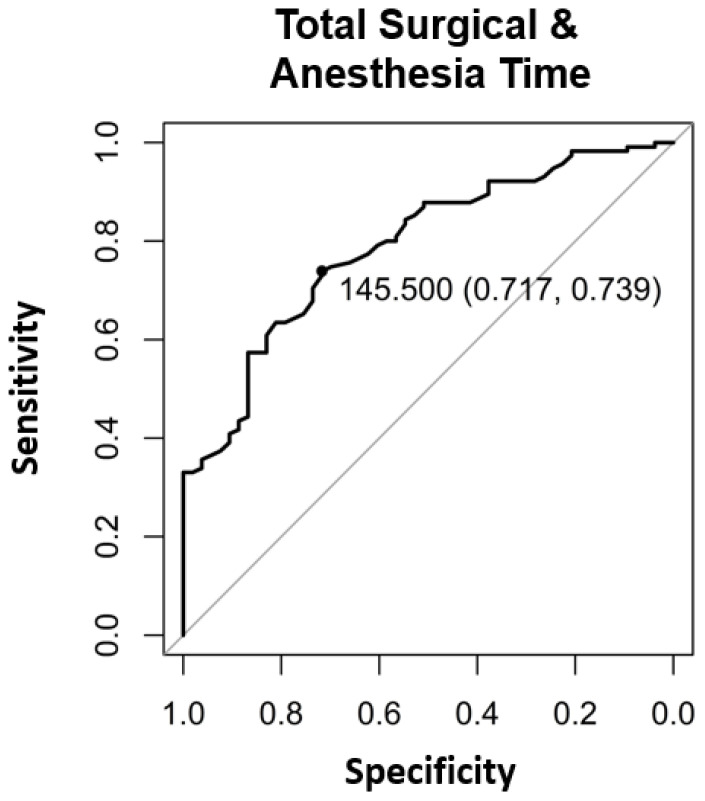
The binary break-even point for total surgical time and anesthesia time was 145.5 min determined with Area under the Curve (AUC) and the “closest-to-(0, 1)-criterion”.

**Figure 4 healthcare-09-00862-f004:**
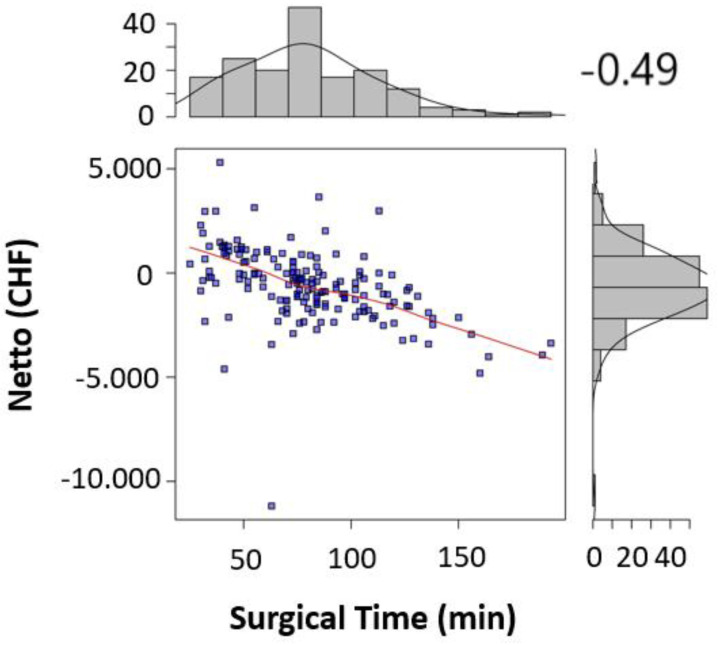
Surgical time was correlated to the contribution margin for each procedure in scatter plots. Sideplots show the distribution of contribution margins and surgical times in a histogram. Strength of correlation was calculated using the Pearson correlation coefficient (*r* = −0.49) and indicated in the figure.

**Figure 5 healthcare-09-00862-f005:**
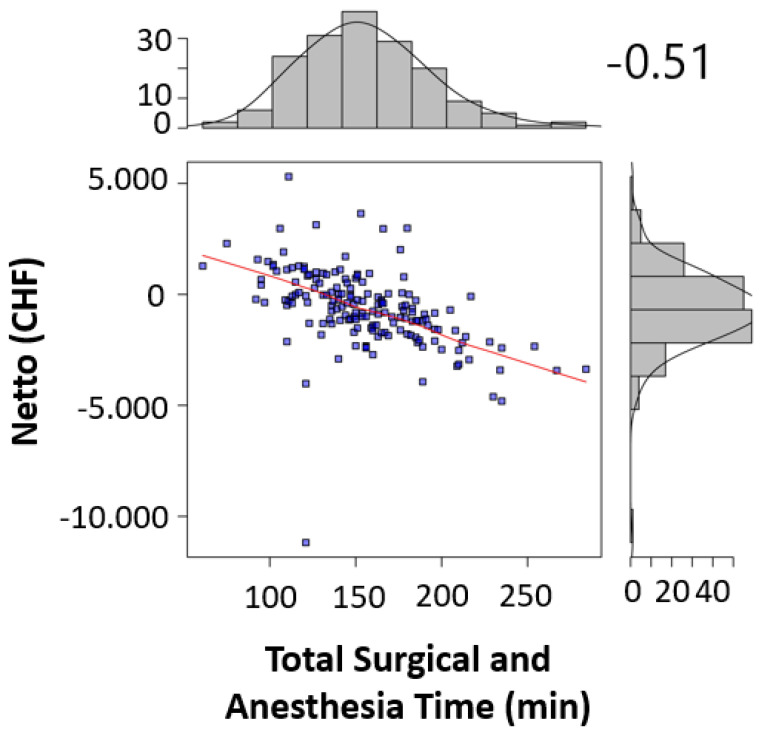
Total surgery and anesthesia time was correlated to the contribution margin for each procedure in scatter plots (A). Sideplots (A) show the distribution of contribution margins and total surgical and anesthesia times in a histogram (A). Strength of correlation was calculated using the Pearson correlation coefficient (*r* = −0.51) and indicated in the figure.

**Figure 6 healthcare-09-00862-f006:**
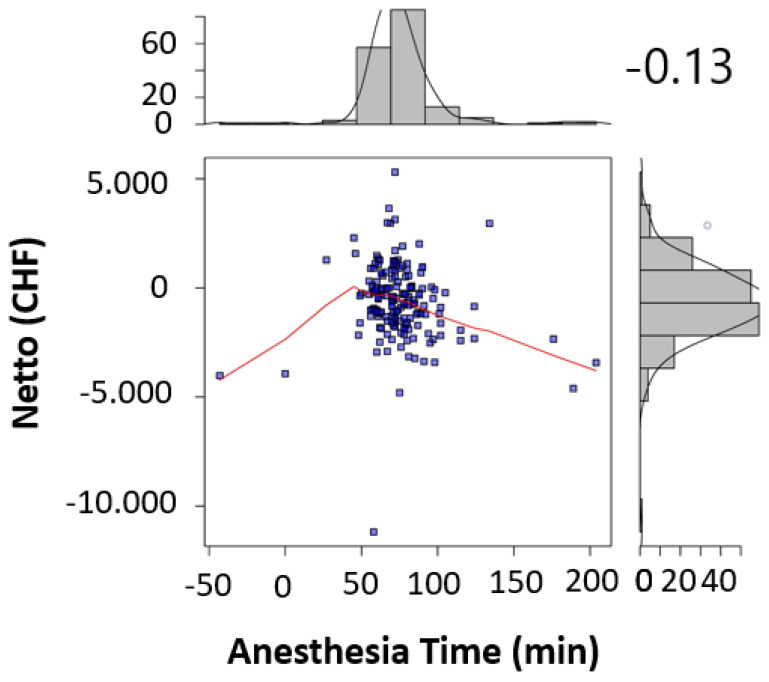
Anesthesia time was correlated to the contribution margin for each procedure in scatter plots (A). Sideplots (A) show the distribution of contribution margins and anesthesia times in a histogram (A). Strength of correlation was calculated using the Pearson correlation coefficient (*r* = −0.13) and indicated in the figure.

**Table 1 healthcare-09-00862-t001:** Variables used for statistical testing with their definition and a formula for continuous variables and parameter values for categorical variables.

Variable	Definition	Formula/Categories
Age	Patient’s age	Date of operation–patient’s date of birth
Sex		Male/female/divers
BMI	Body Mass Index	Patient weight (kg)/(Patient height (cm))^2^
Surgical Time	Period from first surgical cut to final surgical suture	Timepoint (final suture)–timepoint (first surgical cut)
Total Anesthesia Time	Sum of induction time and emergence time as reported by the anesthesiologist	Induction time + Emergence time
ASA-Classification	as defined by the American society of Anesthesiologists	I, II, III, IV, V, VI
Teaching operation	Procedures conducted by or under assistance of residents are considered a *teaching operation*	- Yes, teaching procedure - No, no teaching procedure
Insurance status	Coverage status of patient’s procedure by one of the health care providers shown in the following box	- General Health Insurance (OKP) - Semi-private Health Insurance (SP) - Private Health Insurance (P)
Number of surgeons	Number of surgeons present during the procedure	Numerical value
Unilateral or bilateral operations		- Unilateral - Bilateral
Experience level of the attending operators	Operator’s experience defined by his/her position within the department	- CSu (Chief of Surgery) - CSe (Chief of Service) - AS (Attending Surgeon) - RS (Resident Surgeon)
Additional procedures	Any additional procedures performed during hernia repair surgery	

**Table 2 healthcare-09-00862-t002:** Compared characteristics of patients with a net loss versus those without a net loss.

Parameter	Overall	No Loss	Loss	*p*
Patients (*n*)	168	53	115	
Age (mean/SD)	61.73 (15.82)	63.02 (16.91)	61.13 (15.33)	0.474
Sex = Male	147 (87.5)	45 (84.9)	102 (88.7)	0.660
BMI (mean/SD)	25.09 (3.48)	25.06 (3.69)	25.11 (3.40)	0.939
ASA (mean/SD)	2.12 (0.70)	2.00 (0.71)	2.18 (0.70)	0.118
Prior Surgery	16 (9.5)	4 (7.5)	12 (10.4)	0.757
Insurance-Status				<0.001 *
General—(OKP)	150 (89.3)	40 (75.5)	110 (95.7)	
Supplementary —Semi-private (SP)	14 (8.3)	9 (17.0)	5 (4.3)	
Supplementary —Private (P)	4 (2.4)	4 (7.5)	0 (0.0)	
Bilateral TEP Procedure	115 (68.5)	37 (69.8)	78 (67.8)	0.937
Lichtenstein Procedure	35 (20.2)	4 (7.5)	31 (26.1)	0.010 *
Additional Procedures	18 (10.7)	4 (7.5)	14 (12.2)	0.527
Teaching-Operation	32 (19.0)	3 (5.7)	29 (25.2)	0.005 *
Qualification of Surgeon				0.214
Resident Surgeon (RS)	6 (3.6)	1 (1.9)	5 (4.3)	
Attending Surgeon (AS)	66 (39.3)	16 (30.2)	50 (43.5)	
Chief of Service (CSe)	95 (56.5)	36 (67.9)	59 (51.3)	
Chief of Surgery (CSu)	1 (0.6)	0 (0.0)	1 (0.9)	
Number of Surgeons	2.10 (0.39)	2.00 (0.28)	2.15 (0.42)	0.022 *
Length of Hospital Stay	2.13 (1.05)	2.11 (0.58)	2.14 (1.21)	0.883
Complication	7 (4.2)	0 (0.0)	7 (6.1)	0.156
Surgical time	80.47 (32.14)	60.49 (22.82)	89.68 (31.69)	<0.001 *
Total Anesthesia Time	75.01 (24.39)	70.55 (15.09)	77.06 (27.46)	0.108
Surgical and Anesthesia Time	155.48 (36.98)	131.04 (26.96)	166.74 (35.59)	<0.001 *

* *p* < 0.05, tests used: Welch’s two-sample t test and the Chi-square test.

## Data Availability

The datasets used and/or analyzed during the current study are available from the corresponding author on reasonable request.

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
