# Peer review of "Analysis of Factors Relevant to Revenue Enhancement in Hernia Interventions (SwissDRG G09)"

_healthcare, 2021, doi:10.3390/healthcare9070862_

Round 1

Reviewer 1 Report

Dear authors
This is an interesting study.
The article is well structured and written.

Comments:
Introduction:
- authors may refer to the information in Table 1 in the text, without placing the table itself in the introduction, or they may place it in supplementary material if it is really necessary.

Methods:
- please write the acronym IBM in full
- please clarify the term "educational status of the operation 
- please clarify whether the hospital belongs to the private system or the public system.
- what exactly the insurance status means

3. Results:
In the results please include all the variables described in the methods.

4. The discussion could be improved with a better comparison with other studies and please describe the limitations of the study.

Author Response

Dear Reviewer, please see attachment.

Reviewer 2 Report

Critical review

Abstract:

  • I suggest including the implications of the findings and the main literature contribution of the manuscript. Moreover, add the importance to study this field and the objectives of the paper.

Introduction:

  • It is necessary to divide the paper into (1) introduction and (2) literature review and development of hypothesis.
  • In the introduction part, authors need to include definitions and importance of (1) classification of hernia interventions, (2) revenue enhancement in hernia interventions, (3) coverage of private and public health insurance referring to hernia interventions, and (5) definition of factors relevant, and (6) relationship between factors and revenue enhancement in hernia interventions. All these topics as summary due to each one need to be included in detail in the literature review and development of hypothesis.
  • Moreover, the authors need to introduce (1) the main findings of the research and (2) the main implications (theoretical and practical) of the results.

Literature review and development of hypothesis:

  • The authors need to develop a hypothesis for the study. The hypothesis needs to be grounded in the discussion of previous studies (theoretical and empirical studies) and their results.
  • Authors might include theory and empirical studies (findings) that prove the relationship (positive or negative) between variables.
  • Authors need to discuss the previous studies and their difference from their manuscripts.

Methodology and data:  

  • The authors need to explain better all variables, use references. I suggest to use a table to describe the variable, its definition, and its calculation.
  • I suggest reviewing the structure and modification in (1) research methodology (methodology, definition of variables, formulas, models) and (2) data (source of data and period).
  • The authors need to explain the motivation to use Pearson’s Chi-Square test.

Results:

  • The results need to be aligned with the hypothesis.
  • Prove the hypothesis of the study.
  • Authors need to analyze their findings and their implications in the health, economic and social context.
  • Include the similarities/differences between authors’ results with prior findings.
  • Reorganize the figures and its explanation.
  • Review the spelling of EBITDA in all text.

Conclusions:

  • State the main findings of the research, the contribution of the article, the limitations of the research, and future studies. The conclusions are general, authors need to specify each conclusion or idea.

Round 2

Reviewer 2 Report

The authors modified the manuscript according to reviewer comments.